

# Error-correcting codes for fermionic quantum simulation

Yu-An Chen[1*], Alexey V. Gorshkov[2] and Yijia Xu[2,3†]

**1** International Center for Quantum Materials, School of Physics,
Peking University, Beijing 100871, China
**2** Joint Quantum Institute and Joint Center for Quantum Information and Computer Science,
NIST/University of Maryland, College Park, Maryland 20742, USA
**3** Institute for Physical Science and Technology, University of Maryland,
College Park, Maryland 20742, USA

⋆ yuanchen@pku.edu.cn , † yijia@umd.edu

## Abstract

Utilizing the framework of $\mathbb{Z}_2$ lattice gauge theories in the context of Pauli stabilizer codes, we present methodologies for simulating fermions via qubit systems on a two-dimensional square lattice. We investigate the symplectic automorphisms of the Pauli module over the Laurent polynomial ring. This enables us to systematically increase the code distances of stabilizer codes while fixing the rate between encoded logical fermions and physical qubits. We identify a family of stabilizer codes suitable for fermion simulation, achieving code distances of $d = 2, 3, 4, 5, 6, 7$, allowing correction of any $\lfloor \frac{d-1}{2} \rfloor$-qubit error. In contrast to the traditional code concatenation approach, our method can increase the code distances without decreasing the (fermionic) code rate. In particular, we explicitly show all stabilizers and logical operators for codes with code distances of $d = 3, 4, 5$. We provide syndromes for all Pauli errors and invent a syndrome-matching algorithm to compute code distances numerically.



# 1 Introduction

Error-correcting codes were initially developed to correct quantum errors on noisy quantum devices and have found further applications in condensed matter physics and high-energy physics. The cornerstone of quantum error correction is the stabilizer formalism [1], which defines the codewords in the common eigenspace of elements in an Abelian group, referred to as the stabilizer group. A stabilizer code is labeled $[[n, k, d]]$ when it uses $n$ physical qubits to encode $k$ logical qubits with code distance $d$. The code distance is the minimum weight of an operator that commutes with all elements in the stabilizer group but is not in the stabilizer group itself. The ratios $\frac{k}{n}$ and $\frac{d}{n}$ determine the quality of codes. Recent developments show the existence of "good" quantum low-density parity-check (LDPC) codes, i.e., with the number of logical qubits $k$ and the code distance $d$ both scaling linearly with the number of physical qubits $n$ [2–5]. In this paper, our focus is on a different objective. Instead of encoding logical qubits, we aim to encode logical fermions using physical qubits. This motivation comes from the need to simulate fermions on quantum computers since most models of matter involve electrons, which are, in fact, fermions [6–14]. While fault-tolerant quantum computation [15, 16] is the ultimate goal, current devices still have limited resources and suffer from noise, so error-mitigation schemes are crucial. Therefore, we seek an effective design such that when we implement fermions with qubits on a quantum computer, certain physical qubit errors can be corrected directly in this protocol without having to encode the underlying qubits further. Thus, we want to systematically increase the code distance $d$ in a fermion-to-qubit mapping with a fixed code rate (between logical fermions and physical qubits).[1]

When a fermionic Hamiltonian consists of geometrically local terms, they can be mapped to local qubit operators by the Bravyi-Kitaev superfast encoding and its variants [8, 17], by the auxiliary methods [18–23], or by exact bosonization [24–26]. The mappings between fermionic Hamiltonians and higher-spin Hamiltonians are also studied in Refs. [27–30]. There are also proposals that utilize defects of surface codes for fermionic quantum simulation [31–33], and those defects are recently implemented by Google Quantum AI [34]. Variants of these mappings have been studied to optimize different costs [35–48]. In the context of quantum many-body physics, these mappings also reveal the deep connections between fermion and spin systems [49, 50]. There is another exact fermion-flux lattice duality derived from the $\mathbb{Z}_2$ gauge theory [51]. Aside from the investigation of constructing new mappings, fermion-to-qubit mappings have been studied in the context of variational quantum circuits [52, 53]. In all the above-mentioned methods, extra qubits are required, e.g., the number of qubits is twice the number of fermions on a 2d square lattice. This is the price for the locality-preserving

---

[1]The code rate here is defined as the ratio between the number of logical fermionic modes $k$ and the number of physical qubits $n$, in the $n \to \infty$ limit. Each code will be demonstrated in an infinite plane, but they can be defined on a torus or open disk (up to some boundary modifications) with linear size $L$, such that both $n$ and $k$ scale with $L^2$. If $L$ is sufficiently large, the boundary effects are negligible, and the ratio $k/n$ will converge to the code rate.

property.[2] These methods can be thought of as stabilizer codes. Given $N$ fermions with the Hilbert space dimension $2^N$, they are mapped to $2N$ qubits with space dimension $2^{2N}$, which is an enlarged space. After $N$ gauge constraints (stabilizer conditions) are imposed, the gauge-invariant subspace (code space) has dimension $2^N$, which matches the dimension of the logical fermions. It has been shown that gauge constraints can be utilized for error correction [54], and code distances can be studied for these stabilizer codes. Ref. [17] demonstrates that an improved Bravyi-Kitaev superfast encoding can correct any single-qubit error in a graph where each vertex has degree $d \geq 6$. In Ref. [55], another version of the Bravyi-Kitaev superfast encoding is proposed, called the "Majorana loop stabilizer code," which is designed to have code distance $d = 3$ such that any single-qubit error can be corrected. However, it is not known how to generalize the Bravyi-Kitaev superfast encoding to produce codes with higher and higher code distances. An alternative approach is code concatenation, where logical qubits in error-correcting codes replace physical qubits in the fermion-to-qubit mappings. However, code concatenation will decrease the code rate between logical fermions and physical qubits, increasing the overhead of fermionic simulation. In this work, we present a method that increases the code distances of the fermion-to-qubit mappings while preserving the code rate.

In this paper, we conjugate an existing stabilizer code with a Clifford circuit.[3] This produces a new stabilizer code. Since the new code is obtained via conjugation by a unitary operator, the algebra of the logical operators is preserved. If we choose the circuit wisely, the new stabilizer code will have a larger code distance ($d \geq 3$), compared to the code distance $d = 2$ of the original exact bosonization. To study Clifford circuits systematically, we utilize the Laurent polynomial method introduced in Refs. [56,57] and further extended in Ref. [58], which shows that any Pauli operator can be written as a vector in a symplectic space. For a system with translational symmetry, e.g., the 2d square lattice, the space of Pauli operators becomes a module over a polynomial ring. Furthermore, this polynomial method can be used to formulate the 2d bosonization concisely [59]. The commutation relations of Pauli operators are determined by the symplectic form. Ref. [60] shows that there is a one-to-one correspondence (up to a translation operator on the lattice):

Automorphism of the symplectic form $\Longleftrightarrow$ Clifford circuit on a 2d square lattice.

Therefore, the problem of finding new codes turns into a problem of searching for "good"[4] automorphisms of the Pauli module with the symplectic form, which can be achieved efficiently by exhaustive numerical search.

In this work, we use the Laurent polynomial method to construct bosonizations on a 2d square lattice. Table 1 is the summary of our results. In Section 2, we review the original 2d bosonization method [24] in Section 2.1 and then pictorially construct 2d bosonizations with distances of 3, 4, and 5 in Section 2.2, 2.3, and 2.4, respectively. We review the Laurent polynomial method in Section. 3.1. In Section. 3.2, we describe all these bosonizations within the framework of the Laurent polynomial method. In addition, in Section. 3.3, we describe a computerized method to search for bosonizations. In Appendix A, we discuss the "syndrome matching" method used to compute the code distance of a given bosonization. In Appendix B, we describe the generators of the symplectic group and choose sixteen elementary automorphisms for our numerical search algorithm. In Appendix C, we show the polynomial representations of an automorphism with a distance of 6 and another with a distance of 7.

---

[2]We can apply the Jordan-Wigner transformation on the 2d lattice by choosing a path including all vertices. However, some local fermionic terms will be mapped to long string operators that are highly nonlocal.

[3]A circuit $U$ is Clifford if and only if $UPU^\dagger$ is a product of Pauli matrices for any given Pauli matrix $P$.

[4]Here, "good" refers to symplectic automorphisms that generate bosonizations with higher code distances while preserving locality and code rates.

Table 1: A comparison of codes based on modified exact bosonization to the Bravyi-Kitaev superfast encoding [8] and to the Majorana loop stabilizer code [55] on a 2d square lattice. The $d = 2$ exact bosonization is equivalent to the Bravyi-Kitaev superfast encoding with a specific choice of the ordering of edges [48]. We list the code distance, as well as the weights (after mapping to qubits) of a fermion occupation term (local fermion parity term), of a hopping term, and of a density-density interaction between nearest neighbors. The weights of the stabilizers are also shown.

| | distance | occupation | hopping | interaction | stabilizer |
|---|---|---|---|---|---|
| Bravyi-Kitaev superfast encoding | 2 | 4 | 6 | 6 | 6 |
| Majorana loop stabilizer code | 3 | 3 | 3-4 | 4-6 | 4-10 |
| Exact bosonization ($d = 3$) | 3 | 4 | 3-5 | 6 | 8 |
| Exact bosonization ($d = 4$) | 4 | 6 | 5-6 | 10 | 10 |
| Exact bosonization ($d = 5$) | 5 | 8 | 5-9 | 12-14 | 12 |
| Exact bosonization ($d = 6$) | 6 | 12 | 6-13 | 16-20 | 18 |
| Exact bosonization ($d = 7$) | 7 | 12 | 7-23 | 16-18 | 26 |

## 2 Results

In Section 2.1, we begin by reviewing the original 2d bosonization on a square lattice from Ref. [24]. Then we demonstrate a new way to perform bosonization with code distances of $d = 3$, $d = 4$, and $d = 5$ in Section 2.2, Section 2.3, and Section 2.4, respectively.

### 2.1 Review of the original bosonization

We first describe the Hilbert space in Fig. 1. The elements associated with vertices, edges, and faces will be denoted by $v$, $e$, and $f$, respectively. On each face $f$ of the lattice, we place a single pair of fermionic creation-annihilation operators $c_f, c_f^\dagger$, or equivalently a pair of Majorana fermions $\gamma_f, \gamma_f'$. The even fermionic algebra consists of local observables with trivial fermionic parity, i.e., local observables that commute with the total fermion parity $(-1)^F \equiv \prod_f (-1)^{c_f^\dagger c_f}$ where $F = \sum_f c_f^\dagger c_f$ is the total fermion number.[5] The even algebra is generated by [24]:

---

[5]The even fermionic algebra can also be considered as the algebra of local observables containing an even number of Majorana operators.

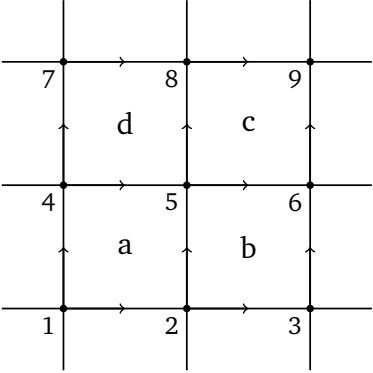

Figure 1: Bosonization on a square lattice [24]. We put Pauli matrices $X_e$, $Y_e$, and $Z_e$ on each edge and one complex fermion $c_f, c_f^\dagger$ on each face. We will work in the Majorana basis $\gamma_f = c_f + c_f^\dagger$ and $\gamma_f' = -i(c_f - c_f^\dagger)$ for convenience.

1. On-site fermion parity (occupation):

$$P_f \equiv -i\gamma_f\gamma_f'.^{[6]}$$  (1)

2. Fermionic hopping term:

$$S_e \equiv i\gamma_{L(e)}\gamma_{R(e)}',$$  (2)

where $L(e)$ and $R(e)$ are faces to the left and right of $e$, with respect to the orientation of $e$ in Fig. 1.

The bosonic dual of this system involves $\mathbb{Z}_2$-valued spins on the edges of the square lattice. For every edge $e$, we define a unitary operator $U_e$ that squares to 1. Labeling the faces and vertices as in Fig. 1, we define:

$$\begin{aligned}
U_{56} &= X_{56}Z_{25}, \\
U_{58} &= X_{58}Z_{45},
\end{aligned}$$  (3)

where $X_e$, $Z_e$ are Pauli matrices acting on a spin at edge $e$:

$$X_e = \begin{bmatrix} 0 & 1 \\ 1 & 0 \end{bmatrix}, \quad Z_e = \begin{bmatrix} 1 & 0 \\ 0 & -1 \end{bmatrix}.$$  (4)

Operators $U_e$ for the other edges are defined by using translation symmetry. Pictorially, operator $U_e$ is depicted as

$$U_e = \quad \overset{\displaystyle \downarrow}{X_e} \quad \text{or} \quad \overset{\displaystyle -X_e-}{\underset{\displaystyle \uparrow}{Z}} $$  (5)
$$-Z-|$$

corresponding to the vertical or horizontal edge $e$.

In Ref. [24], $U_e$ and $S_e$ are shown to satisfy the same commutation relations. We also map the fermion parity $P_f$ at each face $f$ to the "flux operator" $W_f \equiv \prod_{e \subset f} Z_e$, the product of $Z_e$ around a face $f$:

$$W_f = \overset{\displaystyle -Z-}{Z \; f \; Z} $$  (6)
$$-Z-|$$

The bosonization map is

$$\begin{aligned}
S_e &\longleftrightarrow U_e, \\
P_f &\longleftrightarrow W_f,
\end{aligned}$$  (7)

---

[6]$P_f = -i\gamma_f\gamma_f' = (-1)^{c_f^\dagger c_f}$ measures the occupancy of the fermionic mode at face $f$.

or pictorially

$$
\begin{aligned}
i \times \quad & \underset{\gamma'_{R(e)}}{\overset{\gamma_{L(e)}}{\underline{\quad e \quad}}} \quad \longleftrightarrow \quad \overset{-X_e-}{\underset{\qquad}{Z}} \\
\\
i \times \quad & \gamma_{L(e)} \; \Big|\, e \;\; \gamma'_{R(e)} \quad \longleftrightarrow \quad \overset{\qquad}{\underset{-Z-}{X_e}} \\
\\
-i\gamma_f \gamma'_f \quad & \longleftrightarrow \quad \overset{-Z-}{\underset{-Z-}{Z \; f \; Z}}
\end{aligned}
\tag{8}
$$

The condition $P_a P_c S_{58} S_{56} S_{25} S_{45} = 1$ on fermionic operators gives a gauge (stabilizer) constraint $G_v = W_{f_c} \prod_{e \supset v_5} X_e = 1$ for bosonic operators, or generally

$$
G_v = \quad \overset{-Z-}{\underset{X}{\underset{\underset{}{\big|}}{X - v - XZ}}} \; \overset{Z}{\underset{}{}} = 1 .
\tag{9}
$$

The gauge constraint Eq. (9) can be considered as the stabilizer ($G_v |\Psi\rangle = |\Psi\rangle$ for $|\Psi\rangle$ in the code space), which forms the stabilizer group $\mathcal{G}$. The operators $U_e$ and $W_f$ generate all logical operators.[7] The weight of a Pauli string operator $O$ is the number of Pauli matrices in $O$, denoted as wt($O$). For example, we have wt($U_{56}$) = wt($U_{58}$) = 2, wt($W_f$) = 4, and wt($G_v$) = 6. The code distance $d$ is defined as the minimum weight of a logical operator excluding stabilizers:

$$
d = \min\{\text{wt}(O) \mid [O, \mathcal{G}] = 0, \; O \notin \mathcal{G}\} .
\tag{10}
$$

The code distance of this original bosonization is $d = 2$ as $U_e$ has weight 2 and any single Pauli matrix violates at least one $G_v$, which implies that there is no logical operator with weight 1.

There are four types of nearest-neighbor hopping terms ( $\gamma_L \gamma'_R$, $\gamma_L \gamma_R$, $\gamma'_L \gamma'_R$, and $\gamma'_L \gamma_R$) and one type of fermion occupation term ($-i\gamma_f \gamma'_f$). When mapped to Pauli matrices, their weights wt$_i$ are in the range $2 \le \text{wt}_i \le 6$. The maximum weight corresponds to the worst case to simulate the fermion hopping term or the fermion occupation term. A good stabilizer code requires a balance between the minimum weight and the maximum weight. A high minimum weight guarantees the error-correcting property, while a low maximum weight implies that the cost of simulation is low. We label the minimum and maximum weights of the hopping terms as wt$_{\text{min}}$ and wt$_{\text{max}}$. In this example, (wt$_{\text{min}}$, wt$_{\text{max}}$) = (2, 6).

## 2.2 Bosonization with code distance $d = 3$

We now introduce a new way to map the fermionic operators $S_e$ and $P_f$ to Pauli matrices. For simplicity, we present the mapping in a pictorial way:

---

[7]The logical operators consist of all operators that commute with $\mathcal{G}$. Elements of $\mathcal{G}$ are trivial logical operators as stabilizers have no effect on the code space. $U_e$ and $W_f$ together generate all the other logical operators.

$$i \times \quad \begin{array}{c} \gamma_{L(e)} \\ \underline{\quad e \quad} \\ \gamma'_{R(e)} \end{array} \quad \longleftrightarrow \quad \begin{array}{c} Z \\ | \\ -X_e- \\ | \\ Z \end{array}$$

$$i \times \quad \gamma_{L(e)} \ \left| \ e \ \gamma'_{R(e)} \right. \quad \longleftrightarrow \quad \begin{array}{c} X_e \\ | \\ -Z-\!\!-Z- \end{array} \tag{11}$$

$$-i\gamma_f \gamma'_f \quad \longleftrightarrow \quad \begin{array}{c} -Z- \\ Z \ \ f \ \ Z \\ -Z- \end{array}$$

The stabilizer on the bosonic side is

$$G_v^{d=3} = (-1) \times \begin{array}{c} \quad\quad -Z- \\ Z \quad X \quad Z \\ | \quad | \quad | \\ -X- v -X- \\ | \\ X \\ | \\ -Z- \end{array} \quad = 1. \tag{12}$$

Notice that there is a minus sign coming from $ZXZ = -X$.[8] We can manually check that the logical operators defined in Eq. (11) do commute with the stabilizer in Eq. (12). We will prove that this mapping preserves the fermionic algebra in Section. 3.

Given the stabilizer, we can provide the syndromes for all single-qubit Pauli errors, as shown in Fig. 2. We see that all single-qubit Pauli matrices have different syndromes, which means that we do not have any logical operators with weight 2. This implies a code distance of $d \geq 3$. Eq. (11) shows logical operators with weight 3, so we conclude that the code distance is $d = 3$. Based on the syndrome measurements, we can always correct any single-qubit error according to Fig. 2.

The four types of nearest-neighbor hopping terms, $\gamma_L \gamma'_R$, $\gamma_L \gamma_R$, $\gamma'_L \gamma'_R$, and $\gamma'_L \gamma_R$ have weights $\text{wt}_i$ in the range $3 \leq \text{wt}_i \leq 5$. Therefore, the modified bosonization has $(\text{wt}_{\min}, \text{wt}_{\max}) = (3, 5)$ and a fermionic occupation term of weight 4. Compared to the original bosonization with $(\text{wt}_{\min}, \text{wt}_{\max}) = (2, 6)$, the minimum weight is increased such that error correction can be performed, while the maximum weight of the hopping terms is decreased implying a reduction in the simulation cost.

Here we present the spinless Fermi-Hubbard Hamiltonian in 2d square lattice for $d = 3$ encoding as a concrete example. The 2d spinless Fermi-Hubbard Hamiltonian is

$$H = -t \sum_{\langle i,j \rangle} (c_i^\dagger c_j + \text{h.c.}) + U \sum_{\langle i,j \rangle} c_i^\dagger c_i c_j^\dagger c_j, \tag{13}$$

where $\langle i, j \rangle$ represents the nearest-neighbor pair of vertices in the square lattice. We can write individual terms in Majorana basis such as $c_i^\dagger c_j + \text{h.c.} = \frac{i}{2}(\gamma_i \gamma'_j - \gamma'_i \gamma_j)$, $c_i^\dagger c_i c_j^\dagger c_j = (\frac{1 + i\gamma_i \gamma'_i}{2})(\frac{1 + i\gamma_j \gamma'_j}{2})$.

---

[8]The stabilizer is derived from the identity $P_a P_c S_{58} S_{56} S_{25} S_{45} = 1$. After we map $P_f$ and $S_e$ to Pauli matrices using Eq. (11), it becomes $G_v^{d=3}$.

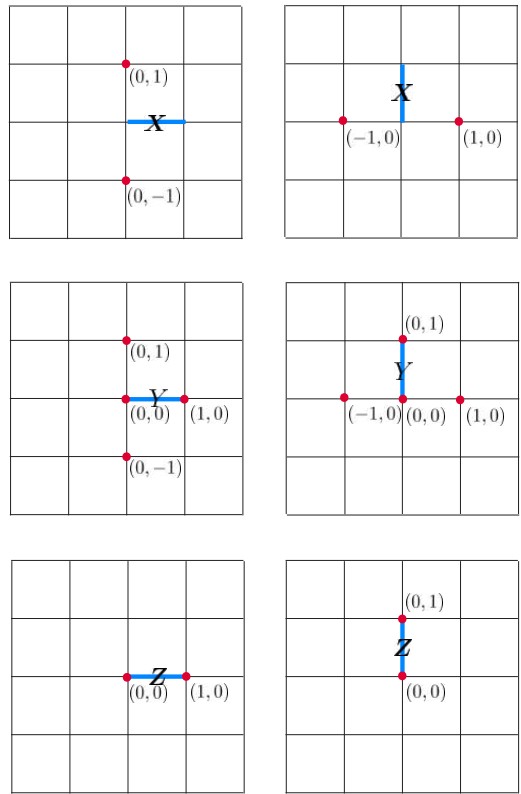

Figure 2: Syndromes of single-qubit errors for the bosonization with code distance $d = 3$. The red vertices $v$ represent locations where the single-qubit error does not commute with the stabilizer $G_v$.

Next, we map the hopping terms and density-density interaction term to Pauli operators as follows

$$
\gamma_i \gamma_j' \longleftrightarrow \quad
\begin{array}{c}
Z \quad i \\
| \\
-X- \\
| \\
Z \quad j
\end{array}
\ , \qquad
\begin{array}{c}
i \quad X \quad j \\
-Z-|-Z-
\end{array}
$$

$$
\gamma_i' \gamma_j \longleftrightarrow \quad
\begin{array}{c}
-Z- \\
i \quad Z \\
-X-| \\
j \quad Z \\
-Z-
\end{array}
\ , \qquad
\begin{array}{c}
-Z- \quad -Z- \\
Z \quad i \quad X \quad j \quad Z
\end{array}
\tag{14}
$$

$$
-\gamma_i \gamma_i' \gamma_j \gamma_j' \longleftrightarrow \quad
\begin{array}{c}
-Z- \\
Z \quad i \quad Z \\
Z \quad j \quad Z \\
-Z-
\end{array}
\ , \qquad
\begin{array}{c}
-Z- \quad -Z- \\
Z \quad i \quad j \quad Z \\
-Z- \quad -Z-
\end{array}
$$

where two terms on the right-hand side correspond to vertical and horizontal $(i, j)$. The hopping terms have weights ranging from 3 to 5, and the interaction term has a weight 6.

## 2.3 Bosonization with code distance $d = 4$

In this subsection, we provide a construction of an exact bosonization with code distance $d = 4$ as an intermediate step toward $d = 5$. Since its code distance is $d = 4$, which is even, this code (like the $d = 3$ code) can only correct the $\lfloor \frac{d-1}{2} \rfloor = 1$ Pauli error. However, for error defection, Pauli errors up to weight 3 can be observed from the stabilizer syndrome measurements.

The mapping can be described as

$$
\begin{aligned}
i \times \ \frac{\gamma_{L(e)}}{\underset{\gamma'_{R(e)}}{e}} &\longleftrightarrow \\
i \times \ \gamma_{L(e)} \ \Big|\ e \ \gamma'_{R(e)} &\longleftrightarrow \\
-i\gamma_f \gamma'_f &\longleftrightarrow
\end{aligned}
\tag{15}
$$

The stabilizer becomes

$$
G^{d=4}_v = \quad = 1 .
\tag{16}
$$

We can check that the logical operators in Eq. (15) commute with the stabilizer in Eq. (16). The proof of the equivalence between the even fermionic algebra and this stabilizer code will be shown in Section. 3.

The syndromes for all single-qubit Pauli errors are provided in Fig. 3. From the generators of the logical operators in Eq. (15), we may be tempted to conclude that the code distance is $d = 5$ because the minimum weight is 5. However, based on the syndromes in Fig. 3, we find that the following operator is logical:

$$
\tag{17}
$$

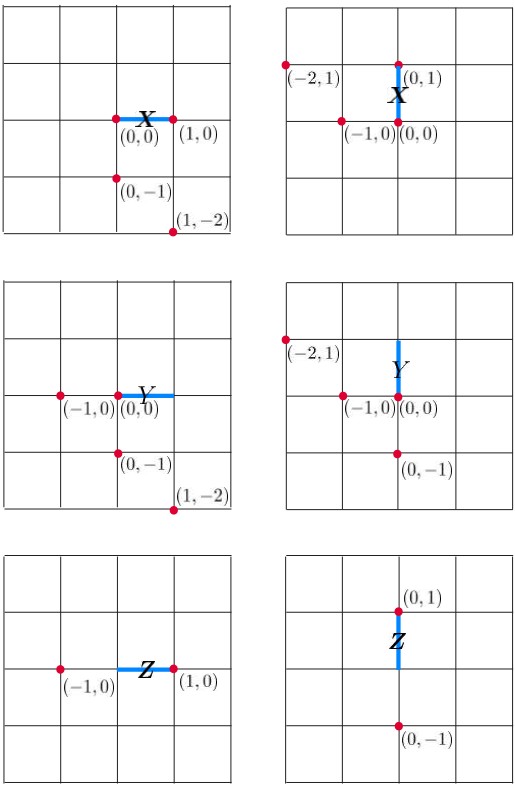

Figure 3: Syndromes of single-qubit errors for the bosonization with code distance $d = 4$. The red vertices $v$ represent locations where the single-qubit error does not commute with the stabilizer $G_v^{d=4}$.

Since it does not commute with the terms in Eq. (15), this operator does not belong to the stabilizer group. Therefore, the code distance for this stabilizer code is $d \leq 4$. In Appendix A, we introduce the "syndrome matching" method to provide a lower bound for the code distance for a given stabilizer code. With this method, we check that the stabilizer code defined by Eq. (16) has a code distance of $d = 4$.

The minimum and maximum weights of the nearest-neighbor terms are $(\text{wt}_{\min}, \text{wt}_{\max}) = (5, 6)$ and the fermionic occupation term has weight 6, which means that all the operations are quite well-balanced. This code has an error-correcting property for any single-qubit error.

## 2.4 Bosonization with code distance $d = 5$

The bosonization map with code distance $d = 5$ is provided in this subsection. The generators of the even fermionic algebra are mapped to Pauli matrices as shown below:

$$
i \quad \frac{\gamma_{L(e)}}{\underset{\gamma'_{R(e)}}{\overset{e}{\rule{2cm}{0.4pt}}}} \quad \longleftrightarrow \quad 
\begin{array}{c}
X \qquad Z \\
\mid\quad\;\, \mid \\
-X_e - \mid - Z - \\
\mid\quad\;\, \mid \\
XZ \qquad Z \\
\mid\quad\;\, \mid \\
-XZ - \mid - Z -
\end{array}
\tag{18}
$$

$$i\gamma_{L(e)}\ \left|\ e\ \ \gamma'_{R(e)}\right. \longleftrightarrow \qquad \left|\ e\qquad \overset{\downarrow}{Z}\right. \tag{19}$$
$$-Z-\ \ -X-\!-X-\!-Z-$$

$$-i\gamma_f\gamma'_f\ \longleftrightarrow \tag{20}$$

The stabilizer is

$$G_v = \qquad\qquad\qquad\qquad = 1\,. \tag{21}$$

The minimum and maximum weights of the nearest-neighbor terms are $(\mathrm{wt}_{\min},\mathrm{wt}_{\max}) = (5,9)$ and the weight of the fermionic occupation term is 8. We use the "syndrome matching" method in Appendix A to confirm that $d = 5$. This code has an error-correcting property for any two-qubit error.

## 3 Stabilizer codes and the Pauli module

This section discusses the stabilizer code formalism and the Pauli module representation via Laurent polynomials. The Laurent polynomial method is reviewed in Section. 3.1. Then, in Section. 3.2, we discuss bosonization with distance $d = 3,4,5$ and the corresponding symplectic automorphisms. The searching algorithm for automorphisms is presented in Section. 3.3.

### 3.1 Review of the Laurent polynomial method for the Pauli algebra

We start by reviewing how any Pauli operator can be expressed as a vector over the **Laurent polynomial ring** $R = \mathbb{F}_2[x,y,x^{-1},y^{-1}]$[9] as set out in Ref. [56]. First, we define $X_{12}, Z_{12}, X_{14}$, and $Z_{14}$ in Fig. 1 as column vectors:

$$X_{12} = \begin{bmatrix} 1 \\ 0 \\ \hline 0 \\ 0 \end{bmatrix}, \quad Z_{12} = \begin{bmatrix} 0 \\ 0 \\ \hline 1 \\ 0 \end{bmatrix}, \quad X_{14} = \begin{bmatrix} 0 \\ 1 \\ \hline 0 \\ 0 \end{bmatrix}, \quad Z_{14} = \begin{bmatrix} 0 \\ 0 \\ \hline 0 \\ 1 \end{bmatrix}. \tag{22}$$

---

[9]This is the ring that consists of all linear combinations of polynomials involving $x$, $x^{-1}$, $y$, $y^{-1}$ (and their powers) with coefficients in $\mathbb{F}_2$ (the field with two elements $\{0,1\}$).

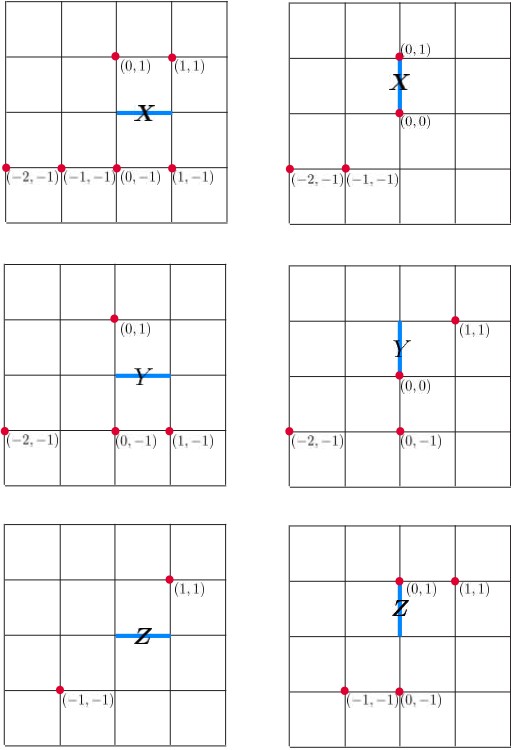

Figure 4: Syndromes of single-qubit errors for the $d = 5$ bosonization.

The Pauli $Y$ can be written as a vector which is a sum of corresponding vectors of $X$ and $Z$, such as

$$Y_{12} = \begin{bmatrix} 1 \\ 0 \\ \hline 1 \\ 0 \end{bmatrix}. \tag{23}$$

We express the vector representation of the Pauli $Y$ operator on an edge $e$ as $XZ$ along the same edge. It is important to note that in this representation, we quotient out the phase factor $\pm 1, \pm i$ associated with Pauli operators. For instance, we equate $Y, -Y, iY, -iY$ with $Y$. Our primary focus is on the commutation and anti-commutation properties of Pauli operators; thus, quotienting out the phase factor $\pm 1, \pm i$ does not impede our calculations. In practical applications, the phase factor can be efficiently tracked, as demonstrated in the Gottesman-Knill theorem [61, 62].

All the other edges can be defined with the help of translation operators as follows. We use polynomials of $x$ and $y$ to represent translation in the $x$ and $y$ directions, respectively. For example,

$$Z_{78} = y^2 \begin{bmatrix} 0 \\ 0 \\ \hline 1 \\ 0 \end{bmatrix} = \begin{bmatrix} 0 \\ 0 \\ \hline y^2 \\ 0 \end{bmatrix}, \quad X_{58} = xy \begin{bmatrix} 0 \\ 1 \\ \hline 0 \\ 0 \end{bmatrix} = \begin{bmatrix} 0 \\ xy \\ \hline 0 \\ 0 \end{bmatrix}. \tag{24}$$

More examples are included in Fig. 5.

Next, we introduce the **antipode map** that is an $\mathbb{F}_2$-linear map from $R$ to $R$ defined by

$$x^a y^b \to \overline{x^a y^b} := x^{-a} y^{-b}. \tag{25}$$

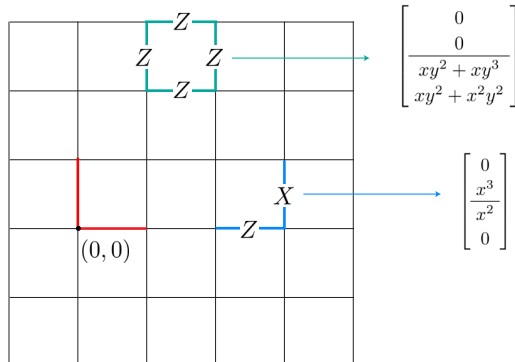

Figure 5: Examples of polynomial expressions for Pauli strings. The flux term (i.e., fermionic occupation) on a plaquette and the hopping term on an edge are both shown. The factors such as $x^2 y^2$ and $x^2$ represent the locations of the operators relative to the origin.

To determine whether two Pauli operators represented by vectors $v_1$ and $v_2$ commute or anti-commute, we define the dot product as

$$v_1 \cdot v_2 = \bar{v}_1^T \Lambda v_2 \,, \tag{26}$$

where $T$ is the transpose operation on a matrix and

$$\Lambda = \left[ \begin{array}{cc|cc} 0 & 0 & 1 & 0 \\ 0 & 0 & 0 & 1 \\ \hline -1 & 0 & 0 & 0 \\ 0 & -1 & 0 & 0 \end{array} \right] \,, \tag{27}$$

is the matrix representation of the standard **symplectic bilinear form**. Notice that $-1$ is the same as 1 because we are working over the $\mathbb{Z}_2$ field. For simplicity, we denote $\overline{(\cdots)}^T$ as $(\cdots)^\dagger$.

The two operators $v_1$ and $v_2$ commute if and only if the constant term of $v_1 \cdot v_2$ is zero. For example, we calculate the dot products

$$X_{12} \cdot Z_{12} = 1 \,, \qquad X_{58} \cdot Z_{14} = x^{-1} y^{-1} \,, \tag{28}$$

and, therefore, $X_{12}$ and $Z_{12}$ anti-commute, whereas $X_{58}$ and $Z_{14}$ commute (their dot product only has a non-constant term $x^{-1} y^{-1}$). Furthermore, the physical meaning of $X_{58} \cdot Z_{14} = x^{-1} y^{-1}$ is that the shifting of $X_{58}$ in $-x$ and $-y$ directions by 1 step will anti-commute with $Z_{14}$. A translationally invariant stabilizer code forms an $R$-submodule[10] $V$ such that

$$v_1 \cdot v_2 = v_1^\dagger \Lambda v_2 = 0 \,, \quad \forall v_1, v_2 \in V \,. \tag{29}$$

We now study the automorphisms $A$ of the symplectic form $\Lambda$:

$$(A v_1) \cdot (A v_2) = v_1 \cdot v_2 \,, \quad \forall v_1, v_2 \in V \,. \tag{30}$$

---

[10]The $R$-submodule is similar to a subspace of a vector space, but the entries of the vector are in the ring $R = \mathbb{F}_2[x, y, x^{-1}, y^{-1}]$. In a ring, the inverse element may not exist. This is the distinction between a module and a vector space.

This is equivalent to $A^\dagger \Lambda A = \Lambda$. All matrices $A$ satisfying this equation form the **symplectic group**. We divide $A$ into $2 \times 2$ blocks $A = \left[\begin{array}{c|c} a & b \\ \hline c & d \end{array}\right]$, the automorphism condition becomes

$$\left[\begin{array}{c|c} a^\dagger & c^\dagger \\ \hline b^\dagger & d^\dagger \end{array}\right]\left[\begin{array}{c|c} 0 & I \\ \hline -I & 0 \end{array}\right]\left[\begin{array}{c|c} a & b \\ \hline c & d \end{array}\right] = \left[\begin{array}{c|c} 0 & I \\ \hline -I & 0 \end{array}\right] \tag{31}$$

$$\Rightarrow \quad a^\dagger d - c^\dagger b = I, \quad a^\dagger c = c^\dagger a, \quad b^\dagger d = d^\dagger b.$$

Examples of the automorphism $A$ are

$$S = \left[\begin{array}{c|c} I & 0 \\ \hline c & I \end{array}\right], \quad \text{where } c \in \text{Mat}_2[R], \text{ and } c^\dagger = c,$$

$$H = \left[\begin{array}{c|c} 0 & I \\ \hline -I & 0 \end{array}\right], \tag{32}$$

$$C = \left[\begin{array}{cc|cc} 1 & 0 & 0 & 0 \\ r & 1 & 0 & 0 \\ \hline 0 & 0 & 1 & \bar{r} \\ 0 & 0 & 0 & 1 \end{array}\right], \quad \text{where } r \in R.$$

$\text{Mat}_2[R]$ consists of all $2 \times 2$ matrices with entries in $R = \mathbb{F}_2[x, y, x^{-1}, y^{-1}]$. The generators of the symplectic group are discussed in Appendix B. We have selected sixteen elementary automorphisms, denoted as $A_1, A_2, \cdots, A_{16}$, from the symplectic group. These automorphisms can be expressed by the conjugation of a unitary operator, i.e., the effect of the automorphism is

$$P \xrightarrow{A} UPU^\dagger. \tag{33}$$

The corresponding unitary circuits for the sixteen elementary automorphisms are illustrated in Fig. 6.

## 3.2 New stabilizer codes developed from automorphisms

First, we reformulate the original bosonization introduced in Section 2.1 and incorporate it into the Pauli module. For simplicity, we will write $x^{-1}$ and $y^{-1}$ as $\bar{x}$ and $\bar{y}$, respectively. The original hopping operators $U_e$ in Eq. (5) can be written as

$$U_1 = \left[\begin{array}{c} 1 \\ 0 \\ \hline 0 \\ \bar{y} \end{array}\right], \quad U_2 = \left[\begin{array}{c} 0 \\ 1 \\ \hline \bar{x} \\ 0 \end{array}\right], \tag{34}$$

where $U_1$ represents $U_e$ on the horizontal edge and $U_2$ represents $U_e$ on the vertical edge. The flux term $W_f$ in (6) is written as

$$W = \left[\begin{array}{c} 0 \\ 0 \\ \hline 1 + y \\ 1 + x \end{array}\right]. \tag{35}$$

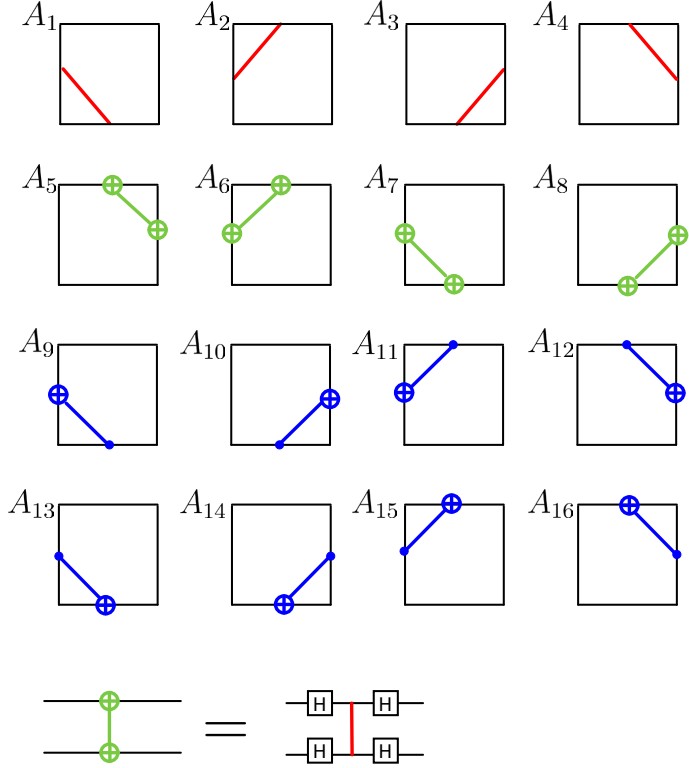

Figure 6: The circuit representations of sixteen elementary automorphisms. Red lines represent the $CZ$ gates, green lines represent the $H^{\otimes 2}(CZ)H^{\otimes 2}$ gates, and blue lines represent the $CNOT$ gates.

The stabilizer $G_v$ in Eq. (9) corresponds to the vector

$$G = \begin{bmatrix} 1+\overline{x} \\ 1+\overline{y} \\ \hline 1+y \\ 1+x \end{bmatrix}. \tag{36}$$

Now, we will apply automorphisms on these vectors to generate new stabilizer codes.

### 3.2.1 Automorphism for code distance $d = 3$

We consider the simplest automorphism

$$A_1 = \left[ \begin{array}{cc|cc} 1 & 0 & 0 & 0 \\ 0 & 1 & 0 & 0 \\ \hline 0 & 1 & 1 & 0 \\ 1 & 0 & 0 & 1 \end{array} \right], \tag{37}$$

which modifies the Pauli operator $X_e$ as

$$A_1 \begin{bmatrix} 1 \\ 0 \\ \hline 0 \\ 0 \end{bmatrix} = \begin{bmatrix} 1 \\ 0 \\ \hline 0 \\ 1 \end{bmatrix}, \quad A_1 \begin{bmatrix} 0 \\ 1 \\ \hline 0 \\ 0 \end{bmatrix} = \begin{bmatrix} 0 \\ 1 \\ \hline 1 \\ 0 \end{bmatrix}. \tag{38}$$

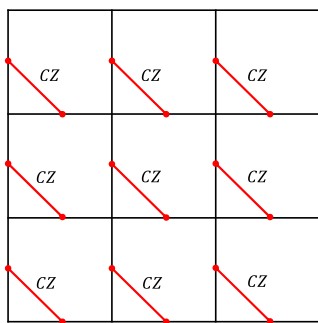

Figure 7: Clifford circuit corresponding to automorphism $A_1$.

Pictorially, this is equivalent to

$$
X_e = \begin{cases} -X_e- & \xrightarrow{A_1} & \overset{\downarrow}{Z} \\[2pt] & & |_{-X_e-} \\[10pt] \overset{\downarrow}{X_e} & \xrightarrow{A_1} & \overset{\downarrow}{X_e} \\[2pt] & & |_{-Z-} \end{cases} \tag{39}
$$

Notice that $Z_e$ is unchanged under this automorphism. This automorphism corrsponds to the Clifford circuit shown in Fig. 7.

Now we apply $A_1$ on the logical operators $U_1$, $U_2$, and $W$ and the stabilizer $G$:

$$
A_1 U_1 = \begin{bmatrix} 1 \\ \hline 0 \\ \hline 0 \\ \hline 1+\overline{y} \end{bmatrix}, \qquad A_1 U_2 = \begin{bmatrix} 0 \\ \hline 1 \\ \hline 1+\overline{x} \\ \hline 0 \end{bmatrix}, \tag{40}
$$

$$
A_1 W = \begin{bmatrix} 0 \\ \hline 0 \\ \hline 1+y \\ \hline 1+x \end{bmatrix}, \qquad A_1 G = \begin{bmatrix} 1+\overline{x} \\ \hline 1+\overline{y} \\ \hline y+\overline{y} \\ \hline x+\overline{x} \end{bmatrix}. \tag{41}
$$

The automorphism $A_1$ applied on $U_e$ can be visualized as

$$
U_e = \begin{cases} \overset{\downarrow}{X_e} & \xrightarrow{A_1} & \overset{\downarrow}{X_e} \\[2pt] -Z-| & & -Z-|_{-Z-} \\[14pt] -X_e- & & \overset{\downarrow}{Z} \\[2pt] \overset{\downarrow}{Z} & \xrightarrow{A_1} & |_{-X_e-} \\[2pt] & & \overset{\downarrow}{Z} \end{cases} \tag{42}
$$

The flux term Eq. (6) is unchanged. The automorphism $A_1$ applied on the stabilizer $G_v$ is

$$G_v = \quad \xrightarrow{\quad A_1 \quad} \tag{43}$$

This is the bosonization with code distance $d = 3$ introduced in Section 2.2. Since we applied the automorphism of the Pauli module on the original bosonization, the logical operators satisfy the same algebra. Therefore, we conclude that this new stabilizer code is a valid way to simulate fermions.

### 3.2.2 Automorphism for code distance $d = 4$

In this section, we consider a slightly more complicated automorphism $A'$:[11]

$$A' = \left[ \begin{array}{cc|cc} 1 & 0 & 0 & 1 \\ 0 & 1 & 1 & 0 \\ \hline 0 & \overline{x}y & 1 + \overline{x}y & 0 \\ x\overline{y} & 0 & 0 & 1 + x\overline{y} \end{array} \right]. \tag{44}$$

One can easily check that $A'$ indeed satisfies the condition in Eq. (31) for being an automorphism. Applying $A'$ on the logical operators $U_1$, $U_2$, and $W$ and the stabilizer $G$, we get

$$A'U_1 = \left[ \begin{array}{c} 1 + \overline{y} \\ 0 \\ \hline 0 \\ x\overline{y}^2 + x\overline{y} + \overline{y} \end{array} \right], \qquad A'U_2 = \left[ \begin{array}{c} 0 \\ 1 + \overline{x} \\ \hline \overline{x} + \overline{x}y + \overline{x}^2 y \\ 0 \end{array} \right], \tag{45}$$

$$A'W = \left[ \begin{array}{c} 1 + x \\ 1 + y \\ \hline 1 + y + \overline{x} + \overline{x}y^2 \\ 1 + x + \overline{y} + \overline{y}x^2 \end{array} \right], \qquad A'G = \left[ \begin{array}{c} x + \overline{x} \\ y + \overline{y} \\ \hline 1 + y + \overline{x} + \overline{x}y^2 \\ 1 + x + \overline{y} + \overline{y}x^2 \end{array} \right]. \tag{46}$$

The operators $A'(U_e)$ can be depicted as

$$U_e = \tag{47}$$

---
[11] $A' = A_4 A_7$ in terms of the elementary automorphisms defined in Appendix B.

The stabilizer $A'(G_v)$ is

$$G_v = \ \substack{-Z-\\XZ\ \ Z\\-X- v\ -XZ-} \quad \xrightarrow{A'} \quad \substack{-Z-\\X\\-Z-\\Z\ \ Z\\-Y- v\ -Z-\ -X-\\Y\ \ Z} \tag{48}$$

The logical operator $U_e$ and the stabilizer $G_v$ are mapped on the bosonization with code distance $d = 4$ introduced in Section 2.3.

Finally, the flux term under the automorphism $A'$ becomes

$$W_f = \ \substack{-Z-\\Z\ \ f\ \ Z\\-Z-} \quad \xrightarrow{A'} \quad \substack{-Z-\\X\\-Z-\ -Z-\\Y\ \ f\ \ Z\\v\ -Y-\ -X-\\Z\ \ Z} \tag{49}$$

This term can be simplified by multiplying it by $A'(G_v)$, which does not change the effective logical operation. The resulting flux term is

$$\substack{-Z-\\X\ \ f\\-Y-\ -X-\\Y\ \ Z} \tag{50}$$

which matches Eq. (15) in Section 2.3.

### 3.2.3  Automorphism for code distance $d = 5$

We introduce another automorphism $A''$:[12]

$$A'' = \left[\begin{array}{cc|cc} 1 & \overline{x} & x & 1 \\ 1 & \overline{x}+1 & 1+x & 1 \\ x & \overline{x}+1 & \overline{x}+1+x+x^2 & 1+x \\ x & 1 & x+x^2 & 1+x \end{array}\right] . \tag{51}$$

---

[12]$A'' = A_9 A_3 A_7 A_{14}$ in terms of the elementary automorphisms defined in Appendix B.

Again it is easy to check that it satisfies the condition in Eq. (31) for being an automorphism. Applying $A''$ on the logical operators $U_1$, $U_2$, and $W$ and the stabilizer $G$, we get

$$A''U_1 = \begin{bmatrix} 1 + \overline{y} \\ \hline 1 + \overline{y} \\ \hline \overline{y} + x\overline{y} + x \\ \overline{y} + x\overline{y} + x \end{bmatrix}, \qquad A''U_2 = \begin{bmatrix} 1 + \overline{x} \\ \hline 0 \\ \hline \overline{x}^2 + x \\ x \end{bmatrix}, \tag{52}$$

$$A''(W + G) = \begin{bmatrix} \overline{xy} + 1 \\ \hline \overline{xy} + \overline{y} \\ \hline \overline{xy} + \overline{y} + \overline{x} + x \\ \overline{y} + x \end{bmatrix}, \qquad A''G = \begin{bmatrix} \overline{xy} + xy \\ \hline \overline{xy} + \overline{y} + y + xy \\ \hline \overline{xy} + \overline{y} + \overline{xy} + y + xy + x^2y \\ \overline{y} + 1 + xy + x^2y \end{bmatrix}. \tag{53}$$

The $A''(U_e)$ operators can be depicted as Eq. (18) and (19). Here, we choose $A''(W + G)$ as our flux operator shown in Eq. (20) because it has a lower weight $\text{wt}[A''(W + G)] < \text{wt}(A''W)$. The pictorial representation of stabilizer $A''G$ is Eq. (21).

## 3.3 Searching algorithm for automorphisms

In this subsection, we describe how we find automorphisms with code distances $d = 3, 4, 5, 6$, and 7. The automorphisms $A_1$ in Eq. (37), $A'$ in Eq. (44), and $A''$ in Eq. (51) correspond to the examples for $d = 3$, $d = 4$, and $d = 5$, respectively. We will show other examples with different code distances.

First, we consider sixteen elementary automorphisms $A_1, A_2, \cdots, A_{16}$ (shown in Appendix B), which attach no more than one new Pauli matrix to the original Pauli matrix. For example, the $A_1$ automorphism attaches one $Z$ to $X_e$, as shown in Eq. (38). Given the sixteen elementary automorphisms, their product $A_{i_1}A_{i_2}A_{i_3}A_{i_4}\cdots$ with $i_n \in \{1, 2, \cdots, 16\}$ is also an automorphism. (We note that these elementary automorphisms do not generate all automorphisms.) We find that the product of five elementary automorphisms $A_{i_1}A_{i_2}A_{i_3}A_{i_4}A_{i_5}$ is sufficient to generate the code distance $d = 7$; therefore, we focus on products with five or fewer elementary automorphisms. We now describe how we search for automorphisms with large code distances:

1. We write down the bosonization of all the nearest-neighbor and on-site terms generated by $S_e$ and $P_f$. They include the automorphism acting on $U_1$, $U_2$, $W$, $U_1 + W$, $U_1 + \overline{y}W$, $U_1 + \overline{y}W + W$, $U_2 + W$, $U_2 + \overline{x}W$, $U_2 + \overline{x}W + W$.[13]

2. We use the minimum weight of bosonization of nearest-neighbor hopping terms to roughly estimate the code distance of a given bosonization (automorphism) $A$.

3. We choose some candidate automorphisms with an appropriate minimum weight $d$, to find a bosonization with desired code distance $d$.

4. We apply the syndrome matching method shown in Appendix A to the candidate automorphisms. By applying syndrome matching, we find a lower bound of their code distances.

5. We apply the syndrome matching method for distance $d + 1$ to an automorphism $\widetilde{A}$ with a lower bound $d$. If syndrome matching for distance $d + 1$ returns a logical operator with no syndrome, we conclude that $\widetilde{A}$ is a bosonization with code distance $d$.

Applying the syndrome-matching method, we found automorphisms to generate exact bosonization with $d = 3, 4, 5, 6, 7$ (see Table 2).

---

[13]The stabilizer $G$ can be added to any term.

Table 2: The possible automorphisms for different code distances. The numbers inside parentheses are the minimum and maximum weights of the logical operators for the nearest-neighbor terms. For example, $A_9A_3A_7A_{14}$ has the minimum logical weight 5 and the maximum logical weight 9.

| Code distance | Automorphisms |
|:---:|:---:|
| 3 | $A_1$ (3,5) |
| 4 | $A_4A_7$ (5,6), $A_2A_7A_1$ (4,6) |
| 5 | $A_9A_3A_7A_{14}$ (5,9) |
| 6 | $A_1A_5A_{14}A_1(6,13)$, $A_4A_9A_{16}A_{11}(7,17)$ |
| 7 | $A_1A_{11}A_5A_{14}A_9(7,23)$ |

## 4 Discussion

This work introduces a method that employs Laurent polynomials and symplectic automorphisms as efficient classical computational tools in the search for fermion-to-qubit mappings with error correction capabilities. One significant advantage of this method lies in its ability to generate equivalent mappings with higher code distances while preserving the code rates. This is possible due to the established equivalence between various 2d fermion-to-qubit mappings, as shown in Ref. [48].

In recent years, it has been shown that Laurent polynomials do not only serve as an analytical tool to design and characterize quantum code [56, 57, 60, 63–65], numerical algorithm inspired by Laurent polynomials are also proposed, for example, searching 3d fracton phases [66]. In this work, we demonstrate the effectiveness of the Laurent polynomial in searching quantum error-correcting codes. The Laurent polynomials and symplectic automorphisms serve a dual purpose in our method: They are instrumental in the analytical derivation of quantum codes and valuable for the numerical studies of these codes. To illustrate the effectiveness of our approach, we demonstrate this method through examples of codes with distances $d = 3, 4, 5$ and further extend our constructions up to $d = 7$. Additionally, we present general algorithms designed to systematically search for and verify fermion-to-qubit mappings with elevated code distances. It is noteworthy that our proposed method is not exclusive to fermion-to-qubit mappings; it is also applicable to regular qubit stabilizer codes, which are utilized to encode logical qubits.

In the context of implementing higher-distance error-correcting codes, which are realized through exact bosonization (with $d = 2$) by Clifford deformations, we can prepare the codewords for these codes. This is achieved by applying the appropriate Clifford circuit to the codewords of the exact bosonization ($d = 2$). The process involves the following steps:

1. Generating a product state in the fermionic Fock basis from the toric code ground state by exciting fermions $\epsilon = e \times m$. This state is a codeword of the exact bosonization ($d = 2$).

2. Transforming this codeword using the Clifford unitary corresponding to the automorphism $A$.

3. The result is the codeword for the higher-distance exact bosonization, preserving the same logical information. In particular, the Clifford circuit used is geometrically local and translationally invariant, ensuring that a shallow, $O(1)$-depth, geometrically local Clifford circuit does not spread errors.

This method provides a framework for developing higher-distance error-correcting codes essential for fermionic quantum simulation, simultaneously underscoring the necessity of efficient decoders for effective error correction. Given that exact bosonization originates from

the toric code, we anticipate that the minimum weight perfect matching approach still works. Apart from quantum error correction, which includes code construction and decoder development, optimizing efficient and fault-tolerant logical operations remains a critical challenge. Exploring efficient decoders and fault-tolerant operations is left for future research.

## Acknowledgment

Y.-A.C wants to thank Zhang Jiang for the discussions that inspired the main idea of this paper. Y.-A.C is also grateful to Nat Tantivasadakarn for teaching the polynomial method and its application to the 2d bosonization. We also offer our thanks to Victor Albert, Riley Chien, Daniel Gottesman, Michael Gullans, Mohammad Hafezi, and Ben Reichardt for engaging in very useful discussions with us.

*Note added.*—As this work was being completed, we also became aware of an independent work, Ref. [67], using a similar polynomial technique to study fermionic quantum simulation from a different perspective. The other recent work Ref. [68] studied an error-mitigation scheme in fermionic encodings. The recent development of programmable neutral atom arrays in Ref. [69] can simulate fermionic modes directly, which avoids using any fermion-to-qubit mapping.

**Funding information**   Y.-A.C. is supported by the JQI fellowship. A.V.G. was supported in part by NSF QLCI (award No. OMA-2120757), DoE ASCR Accelerated Research in Quantum Computing program (award No. DE-SC0020312), DoE QSA, the DoE ASCR Quantum Testbed Pathfinder program (award No. DE-SC0019040), NSF PFCQC program, AFOSR, ARO MURI, AFOSR MURI, and DARPA SAVaNT ADVENT. Y.X. is supported by ARO W911NF-15-1-0397, National Science Foundation QLCI grant OMA-2120757, AFOSR-MURI FA9550-19-1-0399, Department of Energy QSA program. This work is supported by the Laboratory for Physical Sciences through the Condensed Matter Theory Center.

## A   Syndrome-matching method for finding code distances

This appendix presents an algorithm to find the code distance for a given stabilizer code. We call it the "syndrome matching" method. Specifically, given an integer $n$, we describe an algorithm to determine whether the code distance $d$ satisfies $d > n$. The algorithm is as follows.

1. We first choose one vertex to be the origin $(0,0)$. We then apply a single Pauli from $X_1$, $Y_1$, $Z_1$, $X_2$, $Y_2$, $Z_2$ (acting on qubits located on edges $(0,0) \to (1,0)$ and $(0,0) \to (0,1)$). We have 6 cases, each of which has its own syndrome vertices, i.e., a set of vertices $v$ that violate the stabilizer $G_v$. For a single-qubit error, the syndrome set is an ordered set $V^{(1)} = \{(x_1^{(1)}, y_1^{(1)}), (x_2^{(1)}, y_2^{(1)}), ...\}$, ordered by $x_i^{(1)}$: $x_1^{(1)} \leq x_2^{(1)} \leq \ldots$. If two vertices $i, i+1$ have the same $x_i^{(1)} = x_{i+1}^{(1)}$, they are ordered by $y_i^{(1)} < y_{i+1}^{(1)}$.

2. Ensure that the operator is logical. For this to be the case, all syndrome vertices should vanish. Therefore, we select the first syndrome vertex $(x_1^{(k)}, y_1^{(k)}) \in V^{(k)}$. Then we enumerate all choices of a Pauli matrix on an edge different from the Pauli matrix selected in the previous step(s), such that it cancels the syndrome at $(x_1^{(k)}, y_1^{(k)})$. This operation may generate other syndrome vertices. The syndrome vertices $V^{(k)}$ are updated due to this new Pauli matrix. At this stage, the operator has one more Pauli matrix, and a new

ordered set for the syndrome vertices $V^{(k+1)}$. If the syndrome set $V^{(k+1)}$ is empty, this operator is logical. If the operator does not belong to the stabilizer group,[14] this means that we have found the nontrivial logical operator with the minimum weight. This minimum weight is the code distance and, therefore, we stop the algorithm. Otherwise, we continue.

3. Repeat steps 1 and 2 until the algorithm stops automatically or all cases with operators containing $n$ Pauli matrices have been considered, i.e., all $V^{(n)}$ are checked. If the algorithm stops automatically, it will return the value of the code distance. If the algorithm stops by considering all the cases with $n$ Pauli matrices, we conclude that code distance satisfies $d > n$.

# B   Elementary automorphisms

This section demonstrates the transformation rules of Pauli matrices for sixteen elementary automorphisms used in our numerical search algorithm. First, according to Ref. [65], the symplectic group of vectors over the Laurent polynomial ring $R = \mathbb{F}_2[x, y, x^{-1}, y^{-1}]$ is generated by the following $4 \times 4$ matrices, which form the **elementary symplectic group**. Below $a \in R^\times$ is any monomial in $R$.

$$
\begin{aligned}
\text{Hadamard:} \quad & E_{i,i+2}(-1)E_{i+2,i}(1)E_{i,i+2}(-1)\,, & \text{where } 1 \le i \le 2\,, \\
\text{control-Phase:} \quad & E_{i+2,j}(a)E_{j+2,i}(\bar{a})\,, & \text{where } 1 \le i, j \le 2\,, \\
\text{control-Not:} \quad & E_{i,j}(a)E_{j+2,i+2}(-\bar{a})\,, & \text{where } 1 \le i \ne j \le 2\,.
\end{aligned}
\tag{B.1}
$$

where the matrix $E_{i,j}(a)$ for any $a \in R$ is defined as

$$
\left[E_{i,j}(a)\right]_{\mu\nu} = \delta_{\mu\nu} + \delta_{\mu i}\delta_{\nu j}a\,, \quad \text{where } \delta \text{ is the Kronecker delta.}
\tag{B.2}
$$

More explicitly, the Hamamard gates for $i = 1$ and $i = 2$ are

$$
\left[\begin{array}{cc|cc}
0 & 0 & -1 & 0 \\
0 & 1 & 0 & 0 \\
\hline
1 & 0 & 0 & 0 \\
0 & 0 & 0 & 1
\end{array}\right]\,, \quad
\left[\begin{array}{cc|cc}
1 & 0 & 0 & 0 \\
0 & 0 & 0 & -1 \\
\hline
0 & 0 & 1 & 0 \\
0 & 1 & 0 & 0
\end{array}\right]\,.
\tag{B.3}
$$

The control-Phase gates for $(i, j) = (1,1), (1,2), (2,1), (2,2)$ are

$$
\left[\begin{array}{cc|cc}
1 & 0 & 0 & 0 \\
0 & 1 & 0 & 0 \\
\hline
a+\bar{a} & 0 & 1 & 0 \\
0 & 0 & 0 & 1
\end{array}\right]\,, \quad
\left[\begin{array}{cc|cc}
1 & 0 & 0 & 0 \\
0 & 1 & 0 & 0 \\
\hline
0 & a & 1 & 0 \\
\bar{a} & 0 & 0 & 1
\end{array}\right]\,, \quad
\left[\begin{array}{cc|cc}
1 & 0 & 0 & 0 \\
0 & 1 & 0 & 0 \\
\hline
0 & \bar{a} & 1 & 0 \\
a & 0 & 0 & 1
\end{array}\right]\,, \quad
\left[\begin{array}{cc|cc}
1 & 0 & 0 & 0 \\
0 & 1 & 0 & 0 \\
\hline
0 & 0 & 1 & 0 \\
0 & a+\bar{a} & 0 & 1
\end{array}\right]\,.
\tag{B.4}
$$

The control-Not gates for $(i, j) = (1,2), (2,1)$ are

$$
\left[\begin{array}{cc|cc}
1 & a & 0 & 0 \\
0 & 1 & 0 & 0 \\
\hline
0 & 0 & 1 & 0 \\
0 & 0 & -\bar{a} & 1
\end{array}\right]\,, \quad
\left[\begin{array}{cc|cc}
1 & 0 & 0 & 0 \\
a & 1 & 0 & 0 \\
\hline
0 & 0 & 1 & -\bar{a} \\
0 & 0 & 0 & 1
\end{array}\right]\,.
\tag{B.5}
$$

---

[14] A way to check if a Pauli string operator $O$ belongs to the stabilizer group is by computing $O \cdot (AW)$ and $O \cdot (AU_{1,2})$, where $AW$ and $AU_{1,2}$ generate the full logical space since $W$ and $U_{1,2}$ are the original generators in the exact bosonization and we apply an automorphism $A$ on them. $O \cdot (AW) = O \cdot (AU_{1,2}) = 0$ if and only if the operator $O$ commutes with all logical operators, which means $O \in \mathcal{G}$ ($\mathcal{G}$ is the stabilizer group).

From the elementary symplectic group above, we choose the following automorphisms $A_1, ..., A_{16}$, corresponding to applying nearest-neighbor two-qubit Clifford gates.

$$A_1 = \begin{bmatrix} 1 & 0 & 0 & 0 \\ 0 & 1 & 0 & 0 \\ \hline 0 & 1 & 1 & 0 \\ 1 & 0 & 0 & 1 \end{bmatrix}, \qquad A_2 = \begin{bmatrix} 1 & 0 & 0 & 0 \\ 0 & 1 & 0 & 0 \\ \hline 0 & y & 1 & 0 \\ \bar{y} & 0 & 0 & 1 \end{bmatrix}, \tag{B.6}$$

$$A_3 = \begin{bmatrix} 1 & 0 & 0 & 0 \\ 0 & 1 & 0 & 0 \\ \hline 0 & \bar{x} & 1 & 0 \\ x & 0 & 0 & 1 \end{bmatrix}, \qquad A_4 = \begin{bmatrix} 1 & 0 & 0 & 0 \\ 0 & 1 & 0 & 0 \\ \hline 0 & \bar{x}y & 1 & 0 \\ x\bar{y} & 0 & 0 & 1 \end{bmatrix}, \tag{B.7}$$

$$A_5 = \begin{bmatrix} 1 & 0 & 0 & \bar{x}y \\ 0 & 1 & x\bar{y} & 0 \\ \hline 0 & 0 & 1 & 0 \\ 0 & 0 & 0 & 1 \end{bmatrix}, \qquad A_6 = \begin{bmatrix} 1 & 0 & 0 & y \\ 0 & 1 & \bar{y} & 0 \\ \hline 0 & 0 & 1 & 0 \\ 0 & 0 & 0 & 1 \end{bmatrix}, \tag{B.8}$$

$$A_7 = \begin{bmatrix} 1 & 0 & 0 & 1 \\ 0 & 1 & 1 & 0 \\ \hline 0 & 0 & 1 & 0 \\ 0 & 0 & 0 & 1 \end{bmatrix}, \qquad A_8 = \begin{bmatrix} 1 & 0 & 0 & \bar{x} \\ 0 & 1 & x & 0 \\ \hline 0 & 0 & 1 & 0 \\ 0 & 0 & 0 & 1 \end{bmatrix}, \tag{B.9}$$

$$A_9 = \begin{bmatrix} 1 & 0 & 0 & 0 \\ 1 & 1 & 0 & 0 \\ \hline 0 & 0 & 1 & 1 \\ 0 & 0 & 0 & 1 \end{bmatrix}, \qquad A_{10} = \begin{bmatrix} 1 & 0 & 0 & 0 \\ x & 1 & 0 & 0 \\ \hline 0 & 0 & 1 & \bar{x} \\ 0 & 0 & 0 & 1 \end{bmatrix}, \tag{B.10}$$

$$A_{11} = \begin{bmatrix} 1 & 0 & 0 & 0 \\ \bar{y} & 1 & 0 & 0 \\ \hline 0 & 0 & 1 & y \\ 0 & 0 & 0 & 1 \end{bmatrix}, \qquad A_{12} = \begin{bmatrix} 1 & 0 & 0 & 0 \\ x\bar{y} & 1 & 0 & 0 \\ \hline 0 & 0 & 1 & \bar{x}y \\ 0 & 0 & 0 & 1 \end{bmatrix}, \tag{B.11}$$

$$A_{13} = \begin{bmatrix} 1 & 1 & 0 & 0 \\ 0 & 1 & 0 & 0 \\ \hline 0 & 0 & 1 & 0 \\ 0 & 0 & 1 & 1 \end{bmatrix}, \qquad A_{14} = \begin{bmatrix} 1 & \bar{x} & 0 & 0 \\ 0 & 1 & 0 & 0 \\ \hline 0 & 0 & 1 & 0 \\ 0 & 0 & x & 1 \end{bmatrix}, \tag{B.12}$$

$$A_{15} = \begin{bmatrix} 1 & y & 0 & 0 \\ 0 & 1 & 0 & 0 \\ \hline 0 & 0 & 1 & 0 \\ 0 & 0 & \bar{y} & 1 \end{bmatrix}, \qquad A_{16} = \begin{bmatrix} 1 & \bar{x}y & 0 & 0 \\ 0 & 1 & 0 & 0 \\ \hline 0 & 0 & 1 & 0 \\ 0 & 0 & x\bar{y} & 1 \end{bmatrix}. \tag{B.13}$$

The diagonal blocks of $A_1$ are identities, and its nontrivial part is the lower left block which attaches an extra $Z$ to $X_1$ and $X_2$. By multiplying the polynomial vectors of $X_1$, $X_2$, $Z_1$, and $Z_2$ by automorphism $A_1$, we get the following terms:

$$-X_e- \quad \xrightarrow{A_1} \quad \overset{\textstyle Z}{\underset{\textstyle -X_e-}{\big|}} \quad , \qquad \overset{\textstyle X_e}{\big|} \quad \xrightarrow{A_1} \quad \overset{\textstyle X_e}{\underset{\textstyle -Z-}{\big|}}$$

$$-Z_e- \quad \xrightarrow{A_1} \quad -Z_e- \quad , \qquad \overset{\textstyle Z_e}{\big|} \quad \xrightarrow{A_1} \quad \overset{\textstyle Z_e}{\big|} \, . \tag{B.14}$$

Similarly, we may multiply $X_1$, $X_2$, $Z_1$, and $Z_2$ by $A_2$, thereby obtaining

$$
-X_e- \xrightarrow{A_2} \overset{-X_e-}{\underset{|}{\overset{|}{Z}}} \;, \qquad \overset{|}{\underset{|}{X_e}} \xrightarrow{A_2} \overset{-Z-}{\underset{|}{\overset{|}{X_e}}}
$$

$$
-Z_e- \xrightarrow{A_2} -Z_e- \;, \qquad \overset{|}{\underset{|}{Z_e}} \xrightarrow{A_2} \overset{|}{\underset{|}{Z_e}}
$$

(B.15)

Following the same argument, we can obtain pictorial representations for the rest of the automorphisms: $A_3$:

$$
-X_e- \xrightarrow{A_3} \underset{-X_e-|}{\overset{|}{Z}} \;, \qquad \overset{|}{\underset{|}{X_e}} \xrightarrow{A_3} \underset{-Z-|}{\overset{|}{X_e}}
$$

$$
-Z_e- \xrightarrow{A_3} -Z_e- \;, \qquad \overset{|}{\underset{|}{Z_e}} \xrightarrow{A_3} \overset{|}{\underset{|}{Z_e}}
$$

(B.16)

$A_4$:

$$
-X_e- \xrightarrow{A_4} \overset{-X_e-}{\underset{|}{\overset{|}{Z}}} \;, \qquad \overset{|}{\underset{|}{X_e}} \xrightarrow{A_4} \overset{-Z-}{\underset{|}{\overset{|}{X_e}}}
$$

$$
-Z_e- \xrightarrow{A_4} -Z_e- \;, \qquad \overset{|}{\underset{|}{Z_e}} \xrightarrow{A_4} \overset{|}{\underset{|}{Z_e}}
$$

(B.17)

$A_5$:

$$
-X_e- \xrightarrow{A_5} -X_e- \;, \qquad \overset{|}{\underset{|}{X_e}} \xrightarrow{A_5} \overset{|}{\underset{|}{X_e}}
$$

$$
-Z_e- \xrightarrow{A_5} \overset{-Z_e-}{\underset{|}{\overset{|}{X}}} \;, \qquad \overset{|}{\underset{|}{Z_e}} \xrightarrow{A_5} \overset{-X-}{\underset{|}{\overset{|}{Z_e}}}
$$

(B.18)

$A_6$:

$$
-X_e- \xrightarrow{A_6} -X_e- \;, \qquad \overset{|}{\underset{|}{X_e}} \xrightarrow{A_6} \overset{|}{\underset{|}{X_e}}
$$

$$
-Z_e- \xrightarrow{A_6} \underset{X}{\overset{-Z_e-}{\underset{|}{}}} \;, \qquad \overset{|}{\underset{|}{Z_e}} \xrightarrow{A_6} \overset{-X-}{\underset{|}{\overset{|}{Z_e}}}
$$

(B.19)

$A_7$:

$$-X_e- \xrightarrow{A_7} -X_e- \quad , \qquad X_e \xrightarrow{A_7} X_e$$

$$-Z_e- \xrightarrow{A_7} X_{\underset{-Z_e-}{}} \quad , \qquad Z_e \xrightarrow{A_7} Z_e {}_{-X-}$$

(B.20)

$A_8$:

$$-X_e- \xrightarrow{A_8} -X_e- \quad , \qquad X_e \xrightarrow{A_8} X_e$$

$$-Z_e- \xrightarrow{A_8} {}_{-Z_e-}X \quad , \qquad Z_e \xrightarrow{A_8} {}_{-X-}Z_e$$

(B.21)

$A_9$:

$$-X_e- \xrightarrow{A_9} X_{\underset{-X_e-}{}} \quad , \qquad X_e \xrightarrow{A_9} X_e$$

$$-Z_e- \xrightarrow{A_9} -Z_e- \quad , \qquad Z_e \xrightarrow{A_9} Z_e {}_{-Z-}$$

(B.22)

$A_{10}$:

$$-X_e- \xrightarrow{A_{10}} {}_{-X_e-}X \quad , \qquad X_e \xrightarrow{A_{10}} X_e$$

$$-Z_e- \xrightarrow{A_{10}} -Z_e- \quad , \qquad Z_e \xrightarrow{A_{10}} {}_{-Z-}Z_e$$

(B.23)

$A_{11}$:

$$-X_e- \xrightarrow{A_{11}} {}^{-X_e-}X \quad , \qquad X_e \xrightarrow{A_{11}} X_e$$

$$-Z_e- \xrightarrow{A_{11}} -Z_e- \quad , \qquad Z_e \xrightarrow{A_{11}} Z_e {}^{-Z-}$$

(B.24)

$A_{12}$:

$$-X_e- \xrightarrow{A_{12}} \begin{matrix} -X_e- \\ X \\ | \end{matrix} \; , \qquad \begin{matrix} | \\ X_e \\ | \end{matrix} \xrightarrow{A_{12}} \begin{matrix} | \\ X_e \\ | \end{matrix}$$

$$-Z_e- \xrightarrow{A_{12}} -Z_e- \; , \qquad \begin{matrix} | \\ Z_e \\ | \end{matrix} \xrightarrow{A_{12}} \begin{matrix} -Z- \\ Z_e \\ | \end{matrix} \tag{B.25}$$

$A_{13}$:

$$-X_e- \xrightarrow{A_{13}} -X_e- \; , \qquad \begin{matrix} | \\ X_e \\ | \end{matrix} \xrightarrow{A_{13}} \begin{matrix} | \\ X_e \\ -X- \end{matrix}$$

$$-Z_e- \xrightarrow{A_{13}} \begin{matrix} | \\ Z \\ | \\ -Z_e- \end{matrix} \; , \qquad \begin{matrix} | \\ Z_e \\ | \end{matrix} \xrightarrow{A_{13}} \begin{matrix} | \\ Z_e \\ | \end{matrix} \tag{B.26}$$

$A_{14}$:

$$-X_e- \xrightarrow{A_{14}} -X_e- \; , \qquad \begin{matrix} | \\ X_e \\ | \end{matrix} \xrightarrow{A_{14}} \begin{matrix} | \\ X_e \\ -X- \end{matrix}$$

$$-Z_e- \xrightarrow{A_{14}} \begin{matrix} | \\ Z \\ -Z_e- \end{matrix} \; , \qquad \begin{matrix} | \\ Z_e \\ | \end{matrix} \xrightarrow{A_{14}} \begin{matrix} | \\ Z_e \\ | \end{matrix} \tag{B.27}$$

$A_{15}$:

$$-X_e- \xrightarrow{A_{15}} -X_e- \; , \qquad \begin{matrix} | \\ X_e \\ | \end{matrix} \xrightarrow{A_{15}} \begin{matrix} -X- \\ X_e \\ | \end{matrix}$$

$$-Z_e- \xrightarrow{A_{15}} \begin{matrix} -Z_e- \\ Z \\ | \end{matrix} \; , \qquad \begin{matrix} | \\ Z_e \\ | \end{matrix} \xrightarrow{A_{15}} \begin{matrix} | \\ Z_e \\ | \end{matrix} \tag{B.28}$$

$A_{16}$:

$$-X_e- \xrightarrow{A_{16}} -X_e- \; , \qquad \begin{matrix} | \\ X_e \\ | \end{matrix} \xrightarrow{A_{16}} \begin{matrix} -X- \\ X_e \\ | \end{matrix}$$

$$-Z_e- \xrightarrow{A_{16}} \begin{matrix} -Z_e- \\ Z \\ | \end{matrix} \; , \qquad \begin{matrix} | \\ Z_e \\ | \end{matrix} \xrightarrow{A_{16}} \begin{matrix} | \\ Z_e \\ | \end{matrix} \tag{B.29}$$

## C   Automorphisms for code distances $d = 6$ and $d = 7$

In this section, we show the explicit form of automorphisms $A^{d=6}$ and $A^{d=7}$ found by syndrome matching. The automorphism $A^{d=6} = A_1 A_5 A_{14} A_1$ has a code distance of 6:

$$A^{d=6} = \left[ \begin{array}{cc|cc} 1 + \overline{x}y & \overline{x} + y & y & \overline{x}y \\ 0 & 1 + x\overline{y} & x\overline{y} & 0 \\ \hline 0 & x\overline{y} & 1 + x\overline{y} & 0 \\ \overline{x}y & \overline{x} + x + y & x + y & 1 + \overline{x}y \end{array} \right]. \tag{C.1}$$

By applying $A^{d=6}$ on logical operators $U_1$, $U_2$, $W$, and stabilizer $G$, we obtain their polynomial representations as follows:

$$A^{d=6}U_1 = \left[ \begin{array}{c} \overline{x} + 1 + \overline{x}y \\ 0 \\ \hline 0 \\ \overline{y} + \overline{x} + \overline{x}y \end{array} \right], \tag{C.2}$$

$$A^{d=6}U_2 = \left[ \begin{array}{c} \overline{x} + \overline{x}y + y \\ \overline{y} + x\overline{y} + 1 \\ \hline \overline{y} + x\overline{y} + \overline{x} \\ \overline{x} + 1 + x + \overline{x}y + y \end{array} \right], \tag{C.3}$$

$$A^{d=6}W = \left[ \begin{array}{c} \overline{x}y + y^2 \\ x\overline{y} + x \\ \hline x\overline{y} + 1 + x + y \\ 1 + \overline{x}y + xy + y^2 \end{array} \right], \tag{C.4}$$

$$A^{d=6}G = \left[ \begin{array}{c} \overline{x}\overline{y} + \overline{x}^2 y + y + y^2 \\ x\overline{y}^2 + \overline{y} + 1 + x \\ \hline x\overline{y}^2 + 1 + x + y \\ \overline{x}\overline{y} + x\overline{y} + \overline{x} + x + \overline{x}^2 y + y + xy + y^2 \end{array} \right]. \tag{C.5}$$

The automorphism $A^{d=7} = A_1 A_{11} A_5 A_{14} A_9$ with distance $d = 7$ is

$$A^{d=7} = \left[ \begin{array}{cc|cc} \overline{x} + 1 & \overline{x} & y & \overline{x}y + y \\ \overline{x}\overline{y} + \overline{y} + 1 & \overline{x}\overline{y} + 1 & x\overline{y} + 1 & x\overline{y} + \overline{x} + 1 \\ \hline \overline{x}\overline{y} + \overline{y} + 1 & \overline{x}\overline{y} + 1 & x\overline{y} + xy & x\overline{y} + \overline{x} + y + xy \\ \overline{x} + 1 & \overline{x} & x + y & 1 + x + \overline{x}y + y \end{array} \right]. \tag{C.6}$$

By applying $A^{d=7}$ on logical operatrs $U_1$, $U_2$, $W$, and stabilizer $G$, we can write down their polynomial representations as follows:

$$A^{d=7}U_1 = \left[ \begin{array}{c} 0 \\ x\overline{y}^2 + 1 \\ \hline x\overline{y}^2 + \overline{y} + x \\ \overline{y} + x\overline{y} \end{array} \right], \qquad A^{d=7}U_2 = \left[ \begin{array}{c} \overline{x} + \overline{x}y \\ \overline{x}\overline{y} + \overline{y} + \overline{x} + 1 \\ \hline \overline{x}\overline{y} + \overline{y} + 1 + y \\ \overline{x} + 1 + \overline{x}y \end{array} \right], \tag{C.7}$$

$$A^{d=7}W = \left[ \begin{array}{c} \overline{x}y + y + xy + y^2 \\ x^2\overline{y} + \overline{x} + 1 + y \\ \hline x^2\overline{y} + \overline{x} + 1 + x + y + xy + x^2 y + xy^2 \\ 1 + x + x^2 + \overline{x}y + y + y^2 \end{array} \right], \tag{C.8}$$

$$A^{d=7}G = \left[ \begin{array}{c} \overline{xy} + \overline{x}^2 + \overline{x} + 1 + \overline{x}y + y + xy + xy^2 \\ \hline \overline{xy}^2 + \overline{x}^2\overline{y} + \overline{xy} + x\overline{y} + 1 + y \\ \hline \overline{xy}^2 + \overline{x}^2\overline{y} + \overline{xy} + x^2\overline{y} \\ +1 + x + y + xy + x^2y + xy^2 \\ \hline \overline{xy} + \overline{x}^2 + \overline{x} + x + x^2 + \overline{x}y + y + y^2 \end{array} \right] . \tag{C.9}$$

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
