# Peer review of "Error-correcting codes for fermionic quantum simulation"

_SciPost Physics, doi:SciPost Phys. 16, 033 (2024)_

## Round 4 · Referee Report · Anonymous · 2023-12-11

Strengths

1). The authors provide a systematic way of increasing the fermion-qubit code distances while fixing the code rate.
2). The presentation of the background and method is clear.

Weaknesses

1). The discussion of the significance/implications of their findings is limited.

Report

In this paper, the authors introduced a systematic method (Laurent polynomials) to generalize the previously proposed 2D bosonization to a fermion code with a larger code distance while preserving the code rate.
The presentation is clear and their findings are very interesting.
I find their findings useful for the quantum information and computation community, so I recommend their works be published in SciPost Physics.

Requested changes

I have just a few questions/suggestions:

1). Is it possible to represent the Pauli $Y_l$ (or equivalently $X_l Z_l$ with $l$ being a link) operators using the Laurent polynomial vector? It is not very clear to me because $X_l$ and $Z_l$ do not commute.

2). In Section 3, the authors discussed using automorphism ($A$ matrices) to generate higher-distance codes. It would be helpful and more intuitive if they could also present the explicit form of some of the $A$ matrices, namely the corresponding unitary transformations/circuits in terms of the Pauli operators.

3). I find the Discussion section is rather short and it would be great if the authors could elaborate more on connecting their findings to the fault-tolerant fermionic quantum simulation.

---

## Round 4 · Referee Report · Anonymous · 2023-12-27

Strengths

1-Show how to systematically generate higher distance fermion to qubit encodings using very elegant mathematical framework
2- Clear and pedagogical presentation

Weaknesses

1-No discussion of quantum circuit level implementation of the codes.
2-Only the most naive, minimum weight, decoding is mentioned.

Report

The manuscript provides a systematic approach to generating larger code distance encodings of fermions in qubit systems. This is presented elegantly using the Laurent polynomial representation of Pauli operators and automorphisms of the symplectic form. The authors manage to present the quite abstract material in an accessible way , with plenty of illustrative examples and figures. The paper may have benefitted from an extended discussion of how the larger code distance encodings would be implemented in a quantum computer. Even under an assumption of perfect stabilizer measurements, the decoding seems non-trivial given the large codespace. It would also be interesting to see a discussion of how the larger code distance encodings map standard fermion problems, such as the Hubbard model. Nevertheless, I think these are topics that can be left for future study. I recommend that the paper should be accepted to SciPost.

Requested changes

No further revision required

---

## Round 4 · List of Changes

1. We have proofread the text and equations, correcting any typos present.
2. The description of the symplectic group and automorphisms has been elaborated upon. Section 3.1 has been comprehensively rewritten, with the complete list of generators for the symplectic group added to Appendix B.
3. We have refined the introduction to enhance clarity.
4. Additional references related to the topic have been incorporated into the introduction.
5. The manuscript's format has been updated to conform to the guidelines provided by SciPost Physics.

---

## Round 5 · Author Response

Response to ``Report 2 submitted on 2023-12-27 13:42 by Anonymous'':

Strengths 1-Show how to systematically generate higher distance fermion to qubit encodings using very elegant mathematical framework 2- Clear and pedagogical presentation

Weakness 1-No discussion of quantum circuit level implementation of the codes. 2-Only the most naive, minimum weight, decoding is mentioned.

Report The manuscript provides a systematic approach to generating larger code distance encodings of fermions in qubit systems. This is presented elegantly using the Laurent polynomial representation of Pauli operators and automorphisms of the symplectic form. The authors manage to present the quite abstract material in an accessible way , with plenty of illustrative examples and figures. The paper may have benefitted from an extended discussion of how the larger code distance encodings would be implemented in a quantum computer. Even under an assumption of perfect stabilizer measurements, the decoding seems non-trivial given the large codespace. It would also be interesting to see a discussion of how the larger code distance encodings map standard fermion problems, such as the Hubbard model. Nevertheless, I think these are topics that can be left for future study. I recommend that the paper should be accepted to SciPost.

-We thank the referee for their encouraging feedback and valuable suggestions. In response to the comments, we have enriched our manuscript with an additional paragraph in the discussion section, describing the practical implementation of higher-distance exact bosonization on quantum computers. Furthermore, we have incorporated a paragraph that explores the importance of researching decoders and fault-tolerant gate sets as prospective avenues for future work. Given that exact bosonization originates from the toric code, we anticipate that the minimum weight perfect matching approach still works. With respect to the query about the characteristics of the Hubbard Hamiltonian in the context of higher-distance encoding, our manuscript now includes a detailed description of the $d=3$ construction. Specifically, we elucidate that in this construction, the hopping terms are of weight 3~5, while the interaction terms carry a weight of 6.

Response to ``Anonymous Report 1 on 2023-12-11 (Contributed Report)'':

Strengths 1). The authors provide a systematic way of increasing the fermion-qubit code distances while fixing the code rate. 2). The presentation of the background and method is clear.

Weaknesses 1). The discussion of the significance/implications of their findings is limited.

-We appreciate the referee's insightful feedback. In response, we have revised the discussion section to not only revisit previous analytical applications of Laurent polynomials but also to specifically emphasize our discovery of their effectiveness in numerical calculations. This highlights a significant and practical dimension to their use. Additionally, we have expanded the discussion further to explore future directions in fault-tolerant fermionic quantum simulation.

Requested changes I have just a few questions/suggestions: 1). Is it possible to represent the Pauli Y_l (or equivalently X_l Z_l with l being a link) operators using the Laurent polynomial vector? It is not very clear to me because X_l and Z_l do not commute.

-We thank the referee for bringing up this point. Yes, since we only care about either two Pauli operators commuting or anti-commuting with each other and the phase factors +-1 and +-i do not affect the commutation relations, here we use X_l Z_l to represent Y_l, which is equivalent to quotient out the phase factors {1,i} from the Pauli group. The detailed arguments can be found in Ref. [55, 56].

2). In Section 3, the authors discussed using automorphism (A matrices) to generate higher-distance codes. It would be helpful and more intuitive if they could also present the explicit form of some of the A matrices, namely the corresponding unitary transformations/circuits in terms of the Pauli operators.

-We have drawn a figure to explicitly express the unitary circuit corresponding to each automorphism A1~A16. The figure is attached in Section 3.1.

3). I find the Discussion section is rather short and it would be great if the authors could elaborate more on connecting their findings to the fault-tolerant fermionic quantum simulation.

-We are grateful to the referee for highlighting this aspect. In response, we have incorporated discussions on the essential components needed for fault-tolerant fermionic quantum simulation. We acknowledge that the development of efficient decoders and fault-tolerant gate sets is crucial for these simulations. These topics have been identified as areas for future research and investigation.

---

## Round 5 · List of Changes

1. In Section 2.2, we add a couple of paragraphs to discuss the application of d=3 bosonization to the 2d spinless Fermi-Hubbard model.
  2. In Section 3.1, we elaborate on the Laurent polynomial formalism, including the representation of the Pauli Y operator.
  3. In Section 3.1, we add Fig.6, which provides the unitary circuit descriptions of the sixteen elementary automorphisms used in this work.
  4. In Section 4, we elaborate more about the significance and implications of this work. We describe the process of obtaining the codeword of higher-distance encoding from a codeword of the exact bosonization by applying a depth geometrically local and translational-invariant Clifford circuit corresponding to automorphism A. We also discuss the future research directions toward fault-tolerant fermionic quantum simulation.
  5. More references for the previous research about Laurent polynomials and quantum codes were added.
  6. Some typos are fixed.

---

## Editorial Decision

published